# Workplace interventions to prevent suicide: A scoping review

**Nutmeg Hallett** [1]*, **Helen Rees**[2], **Felicity Hannah** [3], **Lorna Hollowood** [1],
**Caroline Bradbury-Jones**[1]

1 School of Nursing and Midwifery, University of Birmingham, Birmingham, United Kingdom, 2 Health and Allied Professionals, Nottingham Trent University, Nottingham, United Kingdom, 3 Queens Medical Centre, Nottingham University Hospitals NHS Trust, Nottingham, United Kingdom

* n.n.hallett@bham.ac.uk

**Data Availability Statement:** All relevant data are within the paper and its Supporting Information files.

**Funding:** NH and HR received funding from NHS England to conduct this review. The funders had no

## Abstract

### Objectives

To map organisational interventions for workplace suicide prevention, identifying the effects, mechanisms, moderators, implementation and economic costs, and how interventions are evaluated.

### Background

Suicide is a devastating event that can have a profound and lasting impact on the individuals and families affected, with the highest rates found among adults of work age. Employers have a legal and ethical responsibility to provide a safe working environment for their employees, which includes addressing the issue of suicide and promoting mental health and well-being.

### Methods

A realist perspective was taken, to identify within organisational suicide prevention interventions, what works, for whom and in what circumstances. Published and unpublished studies in six databases were searched. To extract and map data on the interventions the Effect, Mechanism, Moderator, Implementation, Economic (EMMIE) framework was used. Mechanisms were deductively analysed against Bronfenbrenner's socio-ecological model.

### Results

From 3187 records screened, 46 papers describing 36 interventions within the military, healthcare, the construction industry, emergency services, office workers, veterinary surgeons, the energy sector and higher education. Most mechanisms were aimed at the individual's immediate environment, with the most common being education or training on recognising signs of stress, suicidality or mental illness in oneself. Studies examined the effectiveness of interventions in terms of suicide rates, suicidality or symptoms of mental illness, and changes in perceptions, attitudes or beliefs, with most reporting positive results.

role in the study design, data collection and analysis, decision to publish, or preparation of the manuscript.

**Competing interests:** The authors have declared that no competing interests exist.

Few studies reported economic costs but those that did suggested that the interventions are cost-effective.

## Conclusions

It seems likely that organisational suicide prevention programmes can have a positive impact on attitudes and beliefs towards suicide as well reducing the risk of suicide. Education, to support individuals to recognise the signs and symptoms of stress, mental ill health and suicidality in both themselves and others, is likely to be an effective starting point for successful interventions.

## Introduction

Annually, more than 700,000 die by suicide, with many more making a suicide attempt [1]. Rates of suicide are highest in people of working age; in England suicide is one of the leading causes of death in those aged 20–64 [2]. Suicide is a devastating event that can have a profound and lasting impact on the individuals and families affected, including increasing their risk of suicide [3].

Suicide is a multifaceted phenomenon, which creates challenges to creating prevention strategies due to the complex convergence of risk factors from genetic to psychosocial to cultural [4]. Whilst employment is a protective factor, specific aspects including job pressure and high stress can contribute to increased suicide risk in some occupational groups. For example, national mortality data in the UK showed that between 2011 and 2015 males in low-skilled occupations, particularly construction work, had the highest risks [5]. The same study found that the highest risks in females were in those who were artists; female nurses also had increased risks. Similarly, an examination of the death register in Sweden between 2006 and 2010 found a borderline increased risk of suicide for men in male-dominated professions and women in female-dominated professions [6].

Barriers for the working population in help-seeking behaviour include concerns regarding potential career impact, stigma and confidentiality [7]. Most people who die by suicide do not have formal contact with mental health services [8]. This means that organisations can play a key role in suicide prevention.

Employers have a legal and ethical responsibility to provide a safe working environment for their employees [9]. This includes addressing the issue of suicide and promoting mental health and well-being. Moreover, the loss of an employee to suicide can have a profound impact on the morale and productivity of the remaining staff, leading to increased absenteeism, decreased motivation, and decreased productivity [10]. Therefore, preventing suicide is important for organizations to support the well-being of their employees, fulfil their legal and ethical responsibilities, and maintain a healthy and productive workplace.

Despite the importance of suicide prevention as an organisational activity, the research evidence underpinning such activities is limited. Workplace psychosocial interventions that are aimed at preventing and treating anxiety and depression, particularly those based on cognitive-behavioural therapy, are effective in reducing symptoms [11,12]. As some psychological symptoms, such as low self-esteem, hopelessness and helplessness, as well as major depressive disorder, are risk factors for suicide [13], it is likely that such interventions could play a role in suicide prevention. Suicide risk factors are, however, more wide-ranging than psychological symptoms, and therefore suicide prevention interventions may need to include more than

therapy. The literature on such interventions is growing. In 2008, Takada and Shima [14] identified 16 workplace programmes, and that they included education and training for individuals and managers, developing support networks and cooperation between internal and external resources. In 2017, Witt and colleagues [15] identified 13 studies that focused specifically on interventions aimed at emergency and protective services employees. They found that such interventions reduced suicide rates but identified that further research was needed. To the best of our knowledge, there are no reviews that have examined the contexts by which workplace suicide prevention interventions produce an effect.

The aim of this review was to map organisational interventions for workplace suicide prevention. We sought to i) identify the effects, mechanisms, moderators, implementation and economic costs of suicide prevention interventions, ii) identify how workplace suicide prevention interventions are evaluated and iii) make recommendations for practice and research.

## Methods

### Protocol and registration

The protocol for this review was developed by NH, HR, LH and CBJ and registered with the Open Science Framework (OSF; doi: 10.17605/OSF.IO/9GFRV). The review was conducted in accordance with guidance from the Joanna Briggs Institute [16] and is reported in accordance with the PRISMA statement for reporting scoping reviews [17].

### Conceptual model

The complexity of the workplace suicide prevention interventions means that evaluation lends itself to the realist perspective of identifying what works, for whom and in what circumstances, rather than traditional scientific approaches aimed at identifying effect size within a specified confidence interval [18]. Realism seeks to explain how an intervention works in terms of the interaction between the context, mechanism and outcomes [19], where context refers to the situation around the person, mechanisms are the causal forces that allow an understanding of the relationship between the context and the outcome, and outcomes can be intended and unintended [20].

### Data sources and search strategy

Published and unpublished studies in CINAHL Plus, EMBASE, Ovid Medline, ProQuest including Theses and Dissertations, and Web of Science Core Collection were searched using key terms related to i) suicide and suicide prevention, ii) workplace and iii) intervention. Subject headings were searched where available and searches were combined with Boolean operators, see Table 1. A supplementary search was conducted in Google Scholar with results being extracted using Harzing's Publish or Perish software. Studies were limited to those published in English between January 2002 and August 2022 when the database searching was conducted. Results were uploaded to Rayyan for de-duplication and screening.

**Eligibility criteria.** Eligibility criteria were based on population, concept, context and types of evidence, see Table 2.

### Screening and selection process

At least two reviewers screened each result by title and abstract, with disagreements resolved by discussion with the review team. The remaining full text papers were accessed and screened against the eligibility criteria, again by at least two reviewers with disagreements resolved by discussion.

**Table 1. Search terms by database.**

| Database | Search terms | No. results |
|---|---|---|
| CINAHL Plus | ((MH "Suicide+) OR "suicide" OR (MH "Suicide Prevention (Iowa NIC)") OR "suicide prevention") AND ("workplace" OR "organi#ation" OR (MH "Workforce") OR "workforce") AND (program#" OR "response" OR "strateg#") | 248 |
| EMBASE | (suicide [MH] OR suicide OR "suicide prevention") AND (workplace [MH] OR workplace OR organi?ation OR workforce [MH] OR workforce) AND (intervention OR program? OR response OR strateg?) | 1063 |
| Google Scholar | (suicide OR "suicide prevention") AND (workplace OR organisation OR organization OR workforce) AND (intervention OR program OR programme OR strategy OR strategies) | 980 |
| Ovid MEDLINE | (suicide [MH] OR suicide OR "suicide prevention") AND (workplace [MH] OR workplace OR organi?ation OR workforce [MH] OR workforce) AND (intervention OR program? OR response OR strateg?) | 742 |
| PsycINFO | (suicide [MH] OR suicide OR "suicide prevention") AND (workplace [MH] OR workplace OR organi?ation OR workforce [MH] OR workforce) AND (intervention OR program? OR response OR strateg?) | 387 |
| ProQuest | (noft(suicide OR "suicide prevention")) AND (noft(workplace OR organisation OR organization OR workforce)) AND (noft((intervention OR program OR programme OR strategy OR strategies)) | 500 |
| Web of Science Core Collection | TS = (suicide OR "suicide prevention") AND TS = (workplace OR organisation OR organization OR workforce) AND TS = (intervention OR program OR programme OR strategy OR strategies) | 918 |

## Data charting, extraction and synthesis

Study characteristics were extracted as defined in the protocol (citation, country, organisational context, information about participants, intervention details, and evaluation methods and results if present). Additionally, we categorised context by sector, e.g. military, healthcare etc. To extract and map data on the interventions the Effect, Mechanism, Moderator, Implementation, Economic (EMMIE) framework was used [21], see Table 3. Not only does this work well for highlighting evidence gaps, but it also assists with exploring suicide prevention interventions in terms of what works, for who and in what circumstances, in accordance with realist methods.

Synthesis of the mechanisms of the interventions was conducted in a two-stage process to allow us to synthesise the interventions by type. First, mechanisms were thematically analysed by a process of coding and inductively grouping the mechanisms into themes. Second, deductive analysis was conducted to apply data to Bronfenbrenner's [22] socio-ecological model. Bronfenbrenner's ecological model is a popular theory in the social sciences because it offers a framework through which to examine individuals' relationships within their immediate context, their communities and wider society. In the context of this review, an ecological approach

**Table 2. Eligibility criteria for inclusion in scoping review.**

|  | Inclusion criteria |
|---|---|
| Population | Workforce, i.e., the people who work in a particularl organisation or industry. |
| Concept | Suicide prevention interventions or programmes, i.e., any organisational response, initiative, intervention or strategy aimed at reducing the risk of suicide. |
| Context | Organisations/workplaces |
| Types of evidence | Any primary research; quality improvement; audit. Systematic reviews were used to identify primary research meeting the inclusion criteria. |

**Table 3. Operationalisation of EMMIE (adapted from Johnson et al., 2015 [21]).**

|  | Description | Data to be extracted |
|---|---|---|
| Effect | Overall effect size and direction of effect | Outcome(s) studied, effect size and direction |
| Mechanisms | How the intervention produces its effects | Breakdown of elements of interventions |
| Moderators | Contexts that moderate if mechanisms will be activated to generate the intended effect | Contextual conditions for the intervention |
| Implementation | Barriers and facilitators of the intervention | Documented barriers/facilitators |
| Economic | Is the intervention cost-effective; cost-benefit analysis | Any analysis of costs |

conceptualises suicide as a multifaceted phenomenon, grounded in interplay among personal, situational and socio-cultural factors. We used it to organise the mechanisms of interventions in the results section. This model enabled us to determine the interventions that target suicide prevention at different levels and allows us to consider the interplay between individual factors, and social and environmental influences. Moderators, implementation and economic costs were narratively synthesised by grouping data within and between studies and exploring relationships [23].

## Quality assessment

In accordance with the JBI guidance, no assessment of the quality of the included studies was undertaken [16].

## Results

### Search results

From 3187 records screened, 46 papers describing 36 interventions were included in this scoping review, see Fig 1 [24].

### Characteristics of included studies

Of the included studies, 17 were published between 2002 and 2012 and the remaining 30 were published in the last 10 years, details provided in S1 Table. Geographically the studies were conducted across the globe with 21 in North America, 12 in Australasia, 7 in Europe, 6 in Asia and 1 in Africa. Three studies were conducted in Low- and Middle-income Countries (LMIC; India, South Africa, Ukraine). A variety of sectors were represented: military ($k = 12$), healthcare ($k = 11$), the construction industry ($k = 10$), emergency services (ambulance, fire, police; $k = 8$), office workers ($k = 2$), veterinary surgeons ($k = 2$) and one each in the energy sector and a university.

A minority of studies ($k = 16$) provided data on gender and four studies only included males [25–28]. All but two of the non-healthcare-based studies that described the gender ratio of participants ($k = 6$) included a high male to female participant ratio (73.2%-92.1% male). One military-based study had an almost even split by gender, with 52.4% males [29] and the university-based intervention included 85.6% female participants [30]. Of the 11 interventions aimed at healthcare workers, five did not provide data on gender, and of those that did ($k = 6$) ratios were mixed. More female nurses engaged with HEAR [31]. There were more female students in the pharmacy [32] and medical [33] cohorts. An equal gender mix of medical doctors presented to services aimed at doctors [34,35]. Two studies included suicide rates by gender; in one military study, four out of five suicides were completed by men [36] and in the construction industry all but two of 426 suicides were completed by men [37].

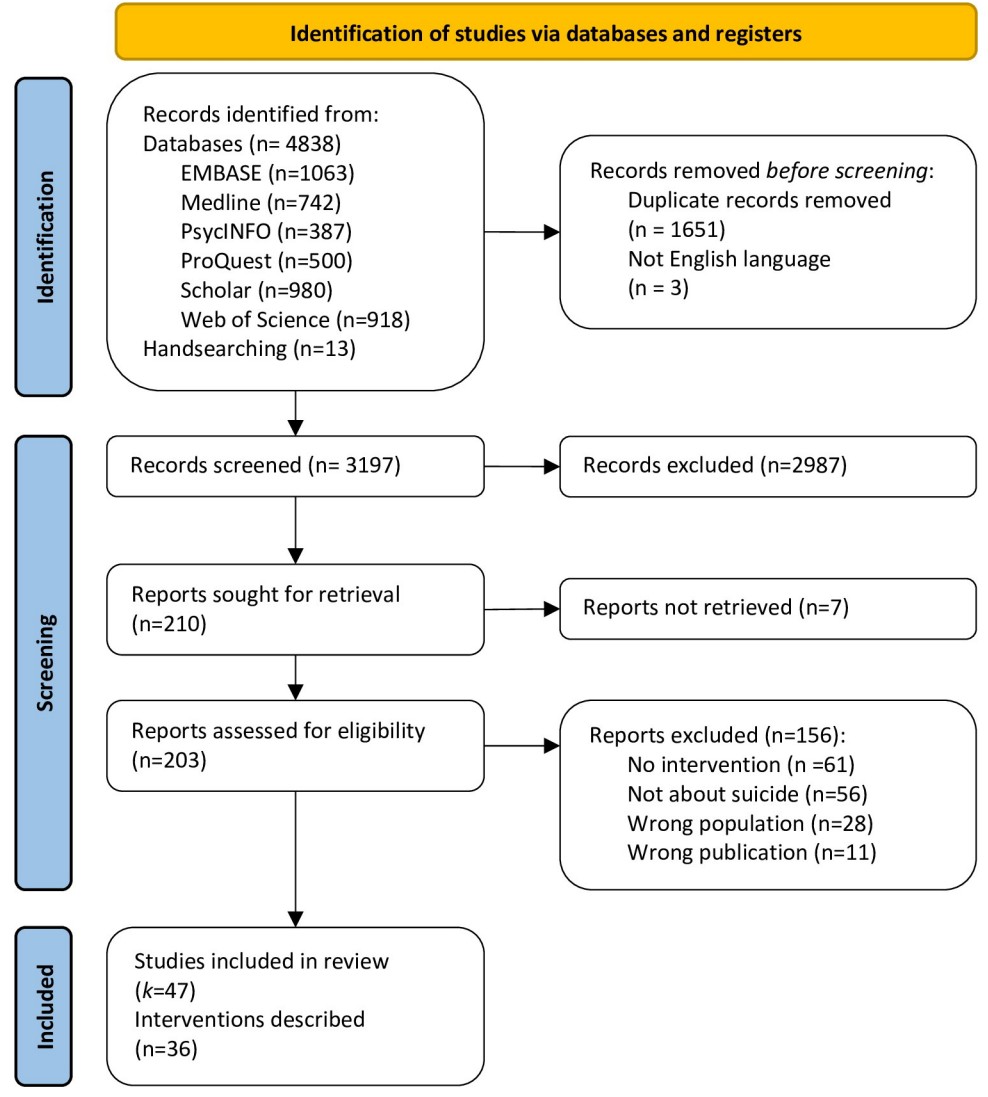

**Fig 1. PRISMA flow diagram [21].**

## Interventions

The 47 included studies described 36 different interventions; details provided in S2 Table. Two interventions were explored across multiple studies. Nine studies examined MATES in Construction, a multi-modal programme aimed at Australian construction workers, either as a whole programme [27,28,37–40] or in part [26,41,42]. One study described a spin-off from this programme, MATES in Energy, aimed at workers in the energy industry [43]. The programme involves general awareness training and training workers to improve mental health and suicide prevention literacy, with the aim of increasing help-seeking behaviours; recruiting volunteer 'connectors' to act as gatekeepers; key-worker suicide first aid 'Applied Suicide Intervention Skills Training' (ASIST); MATES field officers to support workplace volunteers; a support line; and case management for workers at risk of suicide [44]. The (HEAR) programme was developed to increase mental health service utilisation and reduce suicide risk among medical school students [33] and was subsequently adapted for use with nurses [31,45].

HEAR has a two-pronged approach: i) educational training on depression and suicide with the aim of destigmatising mental health treatment and ii) web-based screening, and assessment and referral for those identified at risk [33].

Of the other interventions two were described in initial and follow-up studies: i) the Israeli Defence Force Suicide Prevention Program [29,46] and ii) Together for Life, a suicide prevention programme for the Montreal police [47,48]. One study described three separate interventions [49] and the remaining interventions were described in single studies only.

## Mechanisms of interventions by socio-ecological level

Most mechanisms were aimed at the microsystem, that is, an individual's immediate environment, see Table 4, and this was the case across sectors. The most common mechanism at this level was education or training on recognising signs of stress, suicidality or mental illness in oneself, an element of 18 of the 36 interventions. This was followed by counselling or

**Table 4. Mechanisms of interventions by Socio-Ecological Model (SEM) level.**

| SEM level | Mechanism | Description | Citations |
|---|---|---|---|
| Microsystem (Immediate Environment) | Assessment/screening | Assessment or screening within the intervention for mental health and/or suicide risks. | [27,28,33,36,40,41,47,48,50–58] |
| | Counselling/treatment | Provision of counselling or treatment (medication) within the intervention for mental health and/or suicide risk. | [25,29,33–36,46,50–52,54–56,58–61] |
| | Crisis support | Providing support within the intervention for people at immediate risk of suicide. | [51,53,56,62] |
| | Education for employees (self) | Education or training provided on recognition/management of stress/mental illness/suicidality in oneself. | [25,33,36,49,51,53,54,56–58,62–66] |
| | Employee assistant programme (EAP) | Employee benefit programmes aimed at helping employees deal with personal problems/issues that may affect their performance, health and wellbeing. | [25,60,67] |
| | Health insurance | Provision of employee health insurance for mental health. | [25,60] |
| | Helpline/website | Provision of a helpline or website for people to access support with mental health/suicidality. | [55,59] |
| | Referral/access to external support | Providing referral or access to external support e.g. for counselling/treatment. | [25,33,54,58,60,61,67,68] |
| Mesosystem (Connections) | Family involvement | Providing support to families or enabling family members to be included in the intervention. | [35,36,50] |
| | Gatekeepers | Training or identification of people, from within and without of the organisation, who are strategically to recognise and refer someone at risk of suicide. | [27–29,36,38,40,41,46,49,53,63] |
| | Mentor/supervision | Provision of either mentorship or supervision from managers/ superiors to provide support with, or identify signs/risks of, stress/ mental illness/suicidality. | [52,69] |
| | Peer support/buddy system | Training or identification of peers to provide support with, or identify signs/risks of, stress/mental illness/suicidality. | [27,38,40,52,57,60,61,66,68–70] |
| Exosystem (Indirect Environment) | Education/training for employees [others] or managers | Education or training provided on recognition/management of stress/mental illness/suicidality in others. | [25–30,32,36,38–40,42,43,46–48,51,53,56,61–65,68,69,71,72] |
| | Policy | The introduction/adaption of organisation-wide policies explicitly described in relation to suicide prevention. | [29,36,46,51,52,56] |
| Macrosystem (Social and Cultural Values) | Awareness campaign | Raising awareness of suicidality through leaflets/posters etc. | [28,40] [27,47,48,50,63,64] |
| Chronosystem (Changes Over Time) | Data surveillance | Collection of data on suicides, risk profiles etc. | [36,49,51,56] |
| | Identifying/ monitoring high risk groups | Using data to identify high-risk groups within the workforce, or time points when risks are increased (e.g. on discharge from the military). | [36,51,54,56,60,63] |

treatment (n = 15), and 12 interventions each including assessment or screening and/or referral to external support. The other mechanisms at this level were only seen in 2–5 interventions each.

At the mesosystem level the most common mechanisms were providing a peer support or buddy system (n = 9) and gatekeeper training (n = 6). Only three interventions included family members; the National Mental Health Commission [50] described the interventions available for Australian Defence Force (ADF) members. Family was included in many of the interventions, for example involving family members in supporting service personnel when leaving the ADF, and services aimed specifically at families such as an advocacy service and helpline. Similarly, workshops were delivered for the families and spouses of soldiers transitioning to home after service in the US military [36]. This intervention also included in-service training for spouses aimed at increase soldier resilience. The Villa Sana programme offered one-day or week-long counselling for doctors individually or with their spouse or partner [35].

Training was the most common feature of interventions aimed at the exosystem, i.e. the indirect environment. Eighteen interventions included training for employees or managers aimed at recognising signs of stress, suicidality or mental illness in others. Only three studies described changes to policy, of which two were in the military and one was in healthcare. The Israeli Defence Force suicide prevention programme included a weapon availability reduction policy, ordering soldiers to keep their personal weapons in locked storage when on leave, to reduce availability of means of suicide [46]. The US Air Force introduced an investigative interview policy to ensure that individuals under investigation for legal problems are assessed for risk of suicide [51]. A general hospital in Australia introduced open-door policies for the medical education and medical workforce units to allow for informal two-way communication between junior doctors and management, with the aim of creating a supportive work culture [52].

Awareness campaigns, the only intervention at the macrosystem level, were a feature of four interventions, of which two were aimed at the military. The Ukraine military developed booklets for all soldiers, containing information about suicide and what a soldier can do if they identify another soldier at risk [63]. The Australian Defence Force uses online information resources and e-health services to raise awareness and improve mental health literacy [50]. One of the key facets of MATES in Construction is raising awareness of suicide and mental health through general awareness training, offered to all workers [e.g. 27]. A publicity campaign aimed at the Montreal police force included articles in police newspapers, posters hung in police units and brochures distributed to all members of the force [48].

## Effectiveness of interventions

Interventions were evaluated in fewer than half the studies in this review. In total, 20 studies examined the effectiveness of interventions in terms of suicide rates (n = 10), suicidality or symptoms of mental illness (n = 2), and changes in perceptions, attitudes or beliefs (n = 8). Of the 10 studies that examined suicide rates 9 found a decrease in suicide rates post-intervention implementation, and of these, 7 reported a statistically significant decrease. After implementation of Together for Life in Montreal, there was a 78.9% decrease in suicide rates among Montreal police compared with a non-significant increase for police in Quebec acting as the control [48] and rates remained significantly lower in Montreal when compared with Quebec in the subsequent 10 years [47]. Similarly a significant decrease in rates of suicide was reported in the Serbia and Montenegro [68], Ukrainian [63] and US [56] military post-intervention. Three studies provided risk calculations. An evaluation of MATES found a 9.64% decrease in suicide risk [40]. There was a 33% relative risk reduction in the exposed cohort of the US Air Force [51] and a relative risk of 0.52 post implementation in the Israeli military [29]. After

implementation of IAM awareness training there were no suicides in the follow-up period in the Indian Air Force, however the authors note that rates of suicide were very low prior to implementation [69]. Amongst South African police there was no correlation between the number of suicide prevention workshops delivered and the number of officer suicides by region [62].

Two studies found a reduction in depression symptoms and suicidal ideation following intervention implementation, both conducted in Asia [25,54], although there was no significant reduction of depression as assessed by the Hamilton Depression Scale [25]. Of these, one also found reductions in agitation and guilt [25] and the other in anxiety, insomnia and alcohol use [54].

Improvements in confidence in the application of skills or intervening with a colleague at risk of suicide after training were seen in three studies across industries [27,32,71] whilst increased help-seeking intentions were reported by two evaluations of MATES [27,39]. Shifts towards more favourable attitudes and beliefs about suicide were also reported after MATES with the youngest respondents demonstrating the greatest intervention-associated change [26,42]. Other changes in perceptions were related to improved suicide-literacy [39,42] and perceptions of the safety climate [30].

### Moderators and implementation barriers/facilitators

Few studies identified moderators ($k = 18$), i.e. the contexts that moderate whether a mechanism will be activated to generate an effect, or implementation barriers/facilitators ($k = 6$). There were two main moderators identified. The first was that the intervention should be tailored to the specific context i.e. the police [60] or the construction industry [28,39,40], or that there was an in-house suicide prevention team [64]. Secondly, and perhaps more importantly, was the guarantee that information about suicide assessment or treatment would not appear in personnel records. This was a key moderator for interventions in the military [56], the police [47], and aimed at medics [35] and emergency healthcare workers [61].

Lack of time was identified as a barrier to implementation [49] whilst providing dedicated time was a facilitator [64]. Other barriers were a lack of information about the programme [49], intervention implementation diminishing over time [56] and difficulties in earning the trust of workers [67]. Facilitators of the MATES in Construction programme were promoting the programme on site [27] and being easy to engage with [41].

### Economic costs

Only four studies provided detail about economic costs, two relating to MATES and the others in healthcare. The MATES evaluations use the same figures; authors estimated the total cost of self-harm and suicide in New South Wales, Australia as AU\$527 million in 2010 and estimated savings due to MATES as AU\$3.66 million. The annual budget for MATES was AU\$800,000 and therefore the benefit to cost ratio was 6.4:1, i.e. every AU\$1 invested produced a return of AU\$4.6 [28,40].

Project expenses in the first year of a peer support programme for clinical and non-clinical staff in one healthcare setting were \$12,235 [70]. To set up a wellness programme for medical residents and fellows was estimated to cost \$200,000 [58].

### Discussion

This scoping review has examined global and cross-industry workplace suicide prevention interventions, describing the mechanisms of interventions by socio-ecological level and synthesising the effectiveness, moderators, implementation barriers and facilitators, and

economic costs. Most intervention mechanisms were aimed at a persons' immediate environment and included assessment, treatment, referral and/or crisis support.

Over the past 20 years most studies have been published in the last 10 years. Almost half of the studies in this review were conducted in North America but the remaining studies have global reach, including three LMIC countries. Interventions have been delivered across a variety of sectors, nine in construction and all but one of these related to MATES in Construction. The 12 military studies were across the military (air force, army, special operations, military) whereas 6 of the 11 healthcare studies were aimed at doctors or medical staff. Only three sectors were included in studies that measured effectiveness in terms of rates of suicide: construction, the military and the police.

There have been no randomised controlled trials examining the efficacy of workplace suicide prevention interventions, making it difficult to demonstrate cause and effect. The studies that demonstrated reduced rates of suicide either used before and after study designs or compared rates to other, similar populations. All the studies in this review that did examine rates of suicide found the risk either decreased or rates were lower than comparison populations. This is in line with findings from a review of suicide prevention interventions for emergency and protective services, which found that implementation was associated with suicide rates approximately halving post intervention [15].

This review found that interventions could be successful in changing beliefs and attitudes. There is evidence of the relationship between attitudes towards suicide and suicidal behaviour; having permissive attitudes towards suicide may increase the odds of suicidal ideation, planning and attempts, while viewing suicide as an unjustified behaviour decreases the odds [73]. As demonstrated in this review, attitudes towards suicide can be modified. We did not find any studies that specifically explored the link between modified attitudes and a reduction in suicide behaviours, but evaluations of MATES in Construction have variously reported improvements in attitudes and reduced rates of suicide [e.g. 27,40]. Further research is needed to examine whether modified attitudes directly lead to reduced suicide behaviours.

Even in industries with rates of suicide higher than the general population, suicide is still rare [74]. This means that to truly assess the impact of organisational suicide intervention programmes, we echo the thoughts of Mishara et al. [47 p.188]:

> Researchers who want to directly assess the impact of programs on deaths by suicide either need to have gigantic budgets to evaluate programs with very large numbers of participants, or they must wait many years, while still offering the program, in order to observe a sufficient number of fatalities to have hopes of obtaining significant findings.

Moreover, we believe that more effort needs to be directed at understanding which elements of such programmes, i.e., which mechanisms provide the greatest impact. Otherwise, there is the risk that funds are misdirected away from the most effective mechanisms towards less effective ones. By combining a realist approach with socio-ecological theory we have attempted to begin to examine this but are hampered by the heterogeneity of the interventions and the studies. For example, the interventions that have demonstrated a reduction in risk or rates of suicide had little in common in terms of socio-ecological level addressed (ranging from 2 to 5 levels) nor the mechanisms addressing each level. The only common mechanism within the effective interventions was education either aimed at increasing knowledge for individuals or increasing awareness of signs of stress or suicidality in others. However, it is unclear what additional mechanisms are required to produce a reduction in risk. For example, the IAM training in the Indian Air Force did not have any mechanisms aimed at the microsystem but did include peer support and a mentoring programme, which both addressed the

mesosystem [69]. In comparison, the US Air Force Suicide Prevention Program did not have any mesosystem level interventions but many aimed at the microsystem: assessment, counselling and crisis support [56]. Both programmes were developed for the air force and both demonstrated positive results. What this review cannot state is which elements are most effective or necessary, nor if there are cultural or procedural factors that potentiate one mechanism over another.

The over-representation of a small number of sectors is unsurprising when considered in the context of population suicide prevalence. Suicide is one of the leading causes of death in men aged 20 to 44 [75]. It is this population that tends to make up the construction workforce as well as military personnel [76,77] and whilst there are more women registered as doctors or physicians than who work in construction, they are still in the minority [78]. Hence it is not unexpected that there are more workplace suicide prevention initiatives that are designed for industries where rates of suicide amongst workers are likely to be higher. However, while population rates of suicide among women are lower than men [79], rates of suicide amongst women in nursing are higher than female population suicide rates, globally [8,80,81]. The UK government's suicide prevention strategy highlighted nursing as a high-risk profession for women [82]. Moreover, while the risk of suicide among male medical doctors may be slightly higher than population averages, female doctors' suicide risks are significantly higher [80].

The interventions aimed at healthcare workers in this review had the highest levels of female participants, however none of these studies measured effectiveness in terms of suicide rates and only one identified an improvement in confidence post-intervention [32]. Despite rates of death by suicide being higher among men, women attempt suicide more often; one European study found that two out of three suicide attempt emergency admissions were women [83]. It is important to note that the gender difference may be limited to Western countries; there is little gender difference in rates of suicide in Korea and Japan, for example [84]. The mismatch between rates of suicide attempts and suicide deaths, also known as the 'gender paradox of suicidal behaviour', might be in part because women tend to use less lethal means [85] but other factors, including a higher threshold for help-seeking, may be involved [86]. Increased help-seeking behaviour amongst women appears to be borne out by the findings of this review; the two studies that provided gender data on presentation to services aimed at doctors reported equal numbers of men and women, despite female doctors being in the minority [78]. What is unclear from this review is whether the organisational suicide prevention needs of men and women differ. Suicide risk and protective factors differ by gender, and having an in-depth understanding of the role gender plays in suicidal behaviour is being increasingly recognised as key to improving suicide prevention strategies [87]. Further research is needed to explore organisational suicide prevention activities from a gendered perspective. This may be especially true for high-risk professions such as nursing. Moreover, current research does not factor the impact of intersectionality; in future research it would be beneficial to factor in other characteristics such as ethnicity, sexuality, disability and other protective characteristics in the context of organisational suicide prevention needs.

## Conclusions

It is difficult to draw firm conclusions from this review due to the heterogeneity of the studies we identified. It seems likely however, that organisational suicide prevention programmes can have a positive impact on attitudes and beliefs towards suicide as well reducing the risk of suicide. While we cannot, from the evidence, state with any certainty which elements of interventions are likely to be most effective, we can make some informed suggestions. Education, to support individuals to recognise the signs and symptoms of stress, mental ill health and

suicidality in both themselves and others, is likely to be an effective starting point for successful interventions. Interventions may need to be tailored to the specific context they are to be implemented in. Nearly all the interventions we identified were developed specifically for the sectors that they were implemented in. This is not to say that there is no possibility of transferring interventions between sectors as the MATES programmes have demonstrated; developed in the construction industry, MATES has been successfully adapted for the energy sector. Similarly, the HEAR programme, developed for medical students, was equally as effective in qualified nurses. Finally, interventions need to be trusted and this may be especially true in sectors where disclosure of health concerns could potentially harm the careers or progression of individuals, such as the military.

## Supporting information

**S1 Checklist. Preferred Reporting Items for Systematic reviews and Meta-Analyses extension for Scoping Reviews (PRISMA-ScR) checklist.**
(DOCX)

**S1 File. MEDLINE search strategy.**
(DOCX)

**S1 Table. Study characteristics.**
(DOCX)

**S2 Table. Intervention mechanisms by socioecological level.**
(DOCX)

## Acknowledgments

We appreciate Jordan Omar Blanchard Lafayette's support with title and abstract, and full text screening.

## Author Contributions

**Conceptualization:** Nutmeg Hallett, Helen Rees, Lorna Hollowood, Caroline Bradbury-Jones.

**Data curation:** Nutmeg Hallett, Helen Rees, Felicity Hannah.

**Formal analysis:** Nutmeg Hallett, Felicity Hannah.

**Funding acquisition:** Nutmeg Hallett, Helen Rees, Lorna Hollowood, Caroline Bradbury-Jones.

**Investigation:** Nutmeg Hallett, Helen Rees, Felicity Hannah.

**Methodology:** Nutmeg Hallett, Helen Rees.

**Project administration:** Nutmeg Hallett, Helen Rees.

**Resources:** Nutmeg Hallett, Helen Rees.

**Software:** Nutmeg Hallett.

**Supervision:** Nutmeg Hallett.

**Validation:** Lorna Hollowood, Caroline Bradbury-Jones.

**Writing – original draft:** Nutmeg Hallett, Felicity Hannah.

**Writing – review & editing:** Nutmeg Hallett, Helen Rees, Felicity Hannah, Lorna Hollowood, Caroline Bradbury-Jones.

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
