## [Decision Letter · Decision Letter 0]

31 Oct 2023

PONE-D-23-07980Organisational suicide prevention interventions: A scoping reviewPLOS ONE

Dear Dr. Hallett,

Thank you for submitting your manuscript to PLOS ONE. After careful consideration, we feel that it has merit but does not fully meet PLOS ONE’s publication criteria as it currently stands. Therefore, we invite you to submit a revised version of the manuscript that addresses the points raised during the review process.

We look forward to receiving your revised manuscript.

Kind regards,

Dirceu Henrique Paulo Mabunda

Academic Editor

Reviewers' comments:

Reviewer's Responses to Questions

**Comments to the Author**

1. Is the manuscript technically sound, and do the data support the conclusions?

Reviewer #1: Yes

Reviewer #2: Yes

2. Has the statistical analysis been performed appropriately and rigorously? 

Reviewer #1: N/A

Reviewer #2: N/A

3. Have the authors made all data underlying the findings in their manuscript fully available?

Reviewer #1: Yes

Reviewer #2: Yes

4. Is the manuscript presented in an intelligible fashion and written in standard English?

Reviewer #1: Yes

Reviewer #2: Yes

5. Review Comments to the Author

Reviewer #1: The authors conducted a thorough analysis of workplace suicide prevention interventions in the past 20 years. As the school for children and adolescents, the workplace should represent a key setting for implementing mental health promotion and suicide prevention interventions targeting adults. Nevertheless, research in this field is scarce. Therefore, this scoping review is not only relevant but also stimulating. An added value is that the authors examined the interventions using a realist perspective and tried to identify "what works, for whom and in what circumstances", describing the mechanisms of interventions according to a socio-ecological model.

The article is well well-written and constructed. Only the references need major revision. Furthermore, I would suggest some improvements to make the article easier to read. I included all the suggestions as comments in the attachment.

1. Adding the word "workplace" somewhere in the title could help the article to show up more easily in a search.

2. A few additional citations could be added in the introduction.

3. The in-text citations and the list of references need to be carefully revised. Many studies included in the review are not in the list of references, or their year of publication differs (between the citation in the tables and the reference). References from 43 to 66 are wrong.

4. Supplementary Tables 1 or 2 should be included as main table. Their structure and organization could be improved using a different order when listing the studies (e.g., alphabetical order), adding the reference number and maybe the sample size.

Reviewer #2: This is a very interesting and relevant scoping review of organisational suicide prevention interventions. I just have minor improvements to suggest, as it is well structures and comprehensive. In my opinion, the use of Bronfenbrenner´s ecological model is very appropriate organize the sintesis of results of such a multifactorial problem. My comments are as follow:

a) Consider replacing the word "examine" by "map" in objetive both in abstract (p. 1) and introduction (p. 2)

b) Also in page 2, inform what databases were reviewed to inform that there aren't other reviews that "have examined the contexts by which workplace suicide prevention interventions produce effect".

c) In table 1, the search terms of CINAHL Plus, were not entered.

d) According to JBI methodology you should describe inclusion criteria (Type of participants, Concept, context and types of evidence) "as transparent and unambiguous as possible (JBI, 2020: 431). Please improve table 2 (p. 4)

e) Detail what data was extracted because readers of the paper might not have read the protocol of this scoping review (p.4)

f) The information provided in tables in supplementary data is indispensable to the content of this scoping review and shoud be part of the main text.

g) In page 5, under the heading interventions, mention 47 studies, and everywhere else the number of studies mentioned is 46.

h) In page 9, a reference in missing after the sentence " Even in industries with rates of suicide higher than general population, suicide is still rare.

6. PLOS authors have the option to publish the peer review history of their article (what does this mean?). If published, this will include your full peer review and any attached files.

Reviewer #1: **Yes: **Miriam Iosue

Reviewer #2: No

---

## [Author Response · Author response to Decision Letter 0]

20 Nov 2023

Reviewer comments Authors’ responses

Reviewer #1

The authors conducted a thorough analysis of workplace suicide prevention interventions in the past 20 years. As the school for children and adolescents, the workplace should represent a key setting for implementing mental health promotion and suicide prevention interventions targeting adults. Nevertheless, research in this field is scarce. Therefore, this scoping review is not only relevant but also stimulating. An added value is that the authors examined the interventions using a realist perspective and tried to identify "what works, for whom and in what circumstances", describing the mechanisms of interventions according to a socio-ecological model. 

Thank you for the positive feedback, we appreciate the time you have taken with your review.

The article is well-written and constructed. Only the references need major revision. 

Thank you, I am not sure what happened to the references, they were all correct, but I think something happened with EndNote. It was particularly helpful that you highlighted the incorrect ones. We have gone through and corrected them.

Furthermore, I would suggest some improvements to make the article easier to read. I included all the suggestions as comments in the attachment. The comments in the attachment made it easy to address your comments, we appreciate the time you took to do this.

1. Adding the word "workplace" somewhere in the title could help the article to show up more easily in a search. 

This is a good point, thank you. We have revised the title to ‘Workplace interventions to prevent suicide: A scoping review

.

2. A few additional citations could be added in the introduction. 

We have added detail, with citations, of occupational differences.

3. The in-text citations and the list of references need to be carefully revised. Many studies included in the review are not in the list of references, or their year of publication differs (between the citation in the tables and the reference). References from 43 to 66 are wrong. 

I am not sure what happened but we have carefully gone through the references and they should all now be correct.

4. Supplementary Tables 1 or 2 should be included as main table. Their structure and organization could be improved using a different order when listing the studies (e.g., alphabetical order), adding the reference number and maybe the sample size. 

We agree. While we did not want to include such long tables in the main text, we have included the citations in Table 4. ‘Mechanisms of interventions by Socio-Ecological Model (SEM) level’, which is a summary of Supplementary Table 2. We have not changed the order as it is chronological. We have also not included the sample size due to many of the studies not having methods where a sample size was calculated. 

Reviewer #2: 

This is a very interesting and relevant scoping review of organisational suicide prevention interventions. I just have minor improvements to suggest, as it is well structures and comprehensive. In my opinion, the use of Bronfenbrenner´s ecological model is very appropriate organize the synthesis of results of such a multifactorial problem. 

Thank you, we appreciate the positive feedback.

My comments are as follow:

a) Consider replacing the word "examine" by "map" in objective both in abstract (p. 1) and introduction (p. 2) 

We have replaced ‘examine’ with ‘map’ in the abstract and introduction.

b) Also in page 2, inform what databases were reviewed to inform that there aren't other reviews that "have examined the contexts by which workplace suicide prevention interventions produce effect". 

While we conducted an initial scoping of the literature before we started this review, we did not do this systematically. However, during our systematic database searching we did not uncover any other reviews. It is my understanding that this would not normally be stated in detail in the introduction.

c) In table 1, the search terms of CINAHL Plus, were not entered. 

Thank you, this was an omission, they are now included.

d) According to JBI methodology you should describe inclusion criteria (Type of participants, Concept, context and types of evidence) "as transparent and unambiguous as possible (JBI, 2020: 431). Please improve table 2 (p. 4) 

We have added definitions of workforce and suicide prevention programmes, which we hope will help. When we applied the eligibility criteria, we looked at whether there was a suicide prevention intervention or programme (concept) aimed at the people working in the organisation (population) and provided within the context of the workplace. We believe this is what is stated in the table.

e) Detail what data was extracted because readers of the paper might not have read the protocol of this scoping review (p.4) 

Thank you, we have added detail about the data that were extracted.

f) The information provided in tables in supplementary data is indispensable to the content of this scoping review and should be part of the main text. 

We agree. While we did not want to include such long tables in the main text, we have included the citations in Table 4. ‘Mechanisms of interventions by Socio-Ecological Model (SEM) level’, which is a summary of Supplementary Table 2.

g) In page 5, under the heading interventions, mention 47 studies, and everywhere else the number of studies mentioned is 46.. 

Thank you for picking this typo up, we have corrected it. There were 46 studies.

h) In page 9, a reference in missing after the sentence " Even in industries with rates of suicide higher than general population, suicide is still rare. 

Reference added.

---

## [Decision Letter · Decision Letter 1]

18 Mar 2024

Workplace suicide prevention: A scoping review

PONE-D-23-07980R1

Dear Dr. Hallett,

We’re pleased to inform you that your manuscript has been judged scientifically suitable for publication and will be formally accepted for publication once it meets all outstanding technical requirements.

Kind regards,

Md. Shahjalal

Academic Editor

PLOS ONE

Additional Editor Comments (optional):

Reviewers' comments:

Reviewer's Responses to Questions

**Comments to the Author**

1. If the authors have adequately addressed your comments raised in a previous round of review and you feel that this manuscript is now acceptable for publication, you may indicate that here to bypass the “Comments to the Author” section, enter your conflict of interest statement in the “Confidential to Editor” section, and submit your "Accept" recommendation.

Reviewer #2: All comments have been addressed

Reviewer #3: All comments have been addressed

2. Is the manuscript technically sound, and do the data support the conclusions?

Reviewer #2: Yes

Reviewer #3: Yes

3. Has the statistical analysis been performed appropriately and rigorously? 

Reviewer #2: N/A

Reviewer #3: N/A

4. Have the authors made all data underlying the findings in their manuscript fully available?

Reviewer #2: Yes

Reviewer #3: Yes

5. Is the manuscript presented in an intelligible fashion and written in standard English?

Reviewer #2: Yes

Reviewer #3: Yes

6. Review Comments to the Author

Reviewer #2: I agree that your manuscript improved after the corrections you made. Congratulations. There is a minor correction to be introduced in figure 1, the PRISMA diagram, where it is stated that the number of included studies is 47, not 46.

Reviewer #3: Thank you for the opportunity to review this manuscript. I note that I am reviewing a revision of the initial paper, and also note that the response to the original reviewer/s is satisfactory. This is a well written manuscript covering an important topic. I have no further suggestions and wish you all the best with your paper submission.

7. PLOS authors have the option to publish the peer review history of their article (what does this mean?). If published, this will include your full peer review and any attached files.

Reviewer #2: No

Reviewer #3: No

---

## [Editor Report · Acceptance letter]

24 Apr 2024

PONE-D-23-07980R1 

PLOS ONE

Dear Dr. Hallett, 

I'm pleased to inform you that your manuscript has been deemed suitable for publication in PLOS ONE. Congratulations! Your manuscript is now being handed over to our production team.

Kind regards, 

on behalf of

Dr. Md. Shahjalal 

Academic Editor

PLOS ONE